# JNK Signaling Pathway Involvement in Spinal Cord Neuron Development and Death

**DOI:** 10.3390/cells8121576

**Published:** 2019-12-05

**Authors:** Roberta Schellino, Marina Boido, Alessandro Vercelli

**Affiliations:** 1Department of Neuroscience Rita Levi Montalcini, University of Turin, 10126 Turin, Italy; 2Neuroscience Institute Cavalieri Ottolenghi, University of Turin, 10043 Orbassano (TO), Italy; 3National Institute of Neuroscience (INN), 10125 Turin, Italy

**Keywords:** JNK, spinal cord, motor neurons, MAP kinase

## Abstract

The c-Jun NH2-terminal protein kinase (JNK) is a Janus-faced kinase, which, in the nervous system, plays important roles in a broad range of physiological and pathological processes. Three genes, encoding for 10 JNK isoforms, have been identified: *jnk1*, *jnk2*, and *jnk3*. In the developing spinal cord, JNK proteins control neuronal polarity, axon growth/pathfinding, and programmed cell death; in adulthood they can drive degeneration and regeneration, after pathological insults. Indeed, recent studies have highlighted a role for JNK in motor neuron (MN) diseases, such as amyotrophic lateral sclerosis and spinal muscular atrophy. In this review we discuss how JNK-dependent signaling regulates apparently contradictory functions in the spinal cord, in both the developmental and adult stages. In addition, we examine the evidence that the specific targeting of JNK signaling pathway may represent a promising therapeutic strategy for the treatment of MN diseases.

## 1. Introduction

C-Jun NH2-terminal kinase (JNK) is a member of the mitogen-activated protein kinases (MAPKs): it was discovered more than 20 years ago as the protein kinase family responsible for the transactivation of c-Jun by phosphorylating the N-terminal Ser-63 and Ser-73 residues [1,2]. Since its discovery, a wide amount of studies has been conducted to define the molecular complexity of the JNK signaling pathway in cells. Like the other MAPKs, the activation and increased activity of JNK requires its dual phosphorylation at Tyr and Thr residues, mediated by MAPKKs namely MKK4 and MKK7, which consist of a group of protein kinases with different biochemical properties [3,4]: MKK7 acts as a specific activator of JNK, whereas MKK4 can also activate p38 MAPK; both MKK4 and MKK7 are selectively regulated by extracellular stimuli and show distinct affinity for JNK [4,5]. MKK4 and MKK7 activation is in turn mediated by various MAPKKKs (e.g., apoptosis signal-regulating kinases, ASKs; mixed lineage protein kinases, MLKs; and dual leucine zipper kinase, DLK); moreover the JNK signaling pathway is also modulated by different scaffold proteins, as the JNK-interacting protein (JIP)1, JIP2, and JIP3 [6,7,8].

Three genes, which can be alternatively spliced into 10 isoforms, have been identified: *jnk1*, *jnk2*, and *jnk3* [9]. JNK1 and JNK2 expression is ubiquitous, whereas JNK3 is mainly expressed in the brain, testis and cardiac smooth muscle. Indeed, in most mammalian cell types, JNK activation is strictly controlled and the protein is recruited at moderate levels and at specific circumstances; in contrast, JNK signaling in the brain is highly and constitutively activated [10,11,12,13], thus suggesting that these kinases play a role as key regulator of protein function in the nervous system.

In neurons, JNK signaling is a Janus-faced pathway, which can play a dual role by mediating either physiological or pathological responses [10,13]. Both anti- and pro-apoptotic signals converge on activating MKK4–JNK or MKK7–JNK signaling nodes through specific MAPKKKs, thus contributing to the dual role of this protein family [14]. Moreover, JNKs and the scaffold molecules bind and show distinct patterns of compartmentalization at the subcellular level (nuclear and cytoplasmic), probably related to their pleiotropic function (physiological vs. stress-inducible). Indeed, JNKs are not located statically in a given cellular compartment, but they are able to translocate from the cytoplasm to the nucleus and vice-versa in response to specific stimuli (as for example, stress and excitotoxicity signals released in response to hypoxia or ischemic events [15]. In the nucleus, different transcription factors, such as c-Jun, ATF-2, Elk-1, p53, and NFAT4, which then trigger specific cell stress-responses, are phosphorylated by JNKs [16,17]. In the cytoplasm, interestingly, JNK has been associated with vesicular structures, in particular close to mitochondria [18,19]; indeed, after exposure to excitotoxic stress, JNK is translocated from the nucleus to cytosolic mitochondria, where it can easily phosphorylate those substrates that are known as initiators of programmed cell death following exposure to noxious stimuli [20,21].

While on one side JNK family has been described as involved in injury responses and stress-induced apoptosis in neurodegenerative diseases and, more recently, in the pathophysiology of neuropsychiatric disorders [22,23,24], on the other JNKs are also able to influence neuronal differentiation, by directly targeting chromatin modifiers for modulating histone phosphorylation and acetylation [25]. Therefore, they are involved in the regulation of transcription of those genes related to brain morphogenesis, together with those for axonal growth and pathfinding [18].

In the cytoplasm, evidence show that JNKs are able to activate many non-nuclear substrates, with wide-range functional roles in brain development, such as cell migration, axonal guidance, neurite formation and outgrowth, and also regeneration of nerve fibers after injury [17,26]. Indeed, studies on knockout models of each JNK isoform (Jnk1^−/−^, Jnk2^−/−^, and Jnk3^−/−^) have revealed the role of JNKs in brain development and morphogenesis, together with axodendritic architecture maintenance and restoration [18,27,28,29] (Figure 1).

Thus, the studies performed in these years on JNKs and their molecular pathways identified these molecules as key players in the developing and adult brain, providing the basis for understanding the multifunctional role of JNK signaling in different conditions, cell types, and life stages.

In this review, we will summarize the involvement of the JNK protein family in neuronal physiology and pathology, focusing on the spinal cord. We will describe the role of JNKs in healthy developing and adult nervous system, and in neurodegenerative diseases, with particular attention to those characterized by progressive motor neuron (MN) depletion.

## 2. JNK in CNS Development

Experimental knockout and knockdown approaches in vivo demonstrated that JNKs are functionally active in different stages of brain development: thus, alterations of this molecular pathway can lead to various developmental defects [18]. JNKs are activated following neurulation, when cell proliferation and migration phases are ongoing, and are involved in different developmental processes generally related to axo-dendritic architecture formation and stabilization, and to neuronal pathfinding (Figure 2).

### 2.1. JNK Role in Dendritogenesis

JNK molecules are involved in dendrite architecture regulation and dendrite pruning, which are crucial events for the formation of precise neuronal connections (Figure 2A). Although the specific mechanisms remain to be completely addressed, JNK is thought to act on dendrite morphology by phosphorylating the microtubule-associated proteins 1 and 2 (MAP1 and MAP2) targets, specifically in the dendritic compartment. Indeed, Jnk1–knockout mice show a reduction in phospho-MAP1 and MAP2 in the brain: cerebellar granule neurons isolated from these mice display an increased dendrite complexity, with a higher dendrite number compared to wild-types and reduced dendritic length [18,27,30], suggesting an impairment in microtubule integrity.

Extrinsic factors are also able to phosphorylate JNK signaling, by receptor-mediated activation, to modulate dendritic architecture. For instance, the bone morphogenetic protein receptor II (BMPRII) and the non-canonical WNT pathway activation can increase dendrite complexity via JNK [31,32]. Moreover, the secreted guidance cue semaphorin 3A (Sema3A) acts via the neuropilin 1 receptor and thousand-and one amino acid kinase 2 (TAOK2) molecule to activate JNK signaling at least in cortical neurons [18,33]. Recently, Zhu and colleague identified a novel mechanism involving JNK signaling dendrite pruning in *Drosophila* (c4da) neurons, which are characterized by a drastic remodeling during the metamorphosis: in this model, JNK coordinates with ecdysone receptor B1 (EcRB1) to specifically activate the expression of the downstream Fos protein since the early pupal stage, suggesting that ecdysone signaling provides temporal control of dendritic pruning regulation by JNK signaling. Loss of JNK or its canonical downstream effectors Jun or Fos led to dendrite-pruning defects in c4da neurons. Furthermore, Wnt5 signaling seems to be required for the regulation of dendritogenesis in *Drosophila* neurons and genetically interacts with JNK signaling pathway [34].

### 2.2. JNK Role in Axonogenesis

The activation of JNK is required for axon formation, thus being important for the acquisition of neuron polarity [35]. Indeed, JNK is expressed throughout the neurons, but its phosphorylated form is highly enriched in axons, where it is expressed in a distally directed gradient of increasing intensity, until the growth cone. During axonogenesis, the expression of phospho-JNK in the developing axon is increased about seven folds above the average intensity of the minor processes [35] and this increased phosphorylation reaches the highest level when the axon surpasses the “critical length”, i.e., the extent that unequivocally identifies a process as an axon, compared to the other neurites [36]: this suggests that JNK activation is an early event in axon formation. Transection (e.g., axotomy) proximal to the critical length leads to the loss of neuronal polarity and the possible formation of a new axon from a different site; conversely, when the axon is transected beyond the critical length, it is able to regenerate and maintain the polarity. Thus, it is possible that transection leading to the polarity loss would reduce or eliminate the activated JNK proteins present in the axon [37]. Treatment with specific JNK inhibitors, or with dominant-negative constructs targeting JNK, are able to reversibly but selectively inhibit axonogenesis, because they do not prevent the formation of the minor/shorter processes and their differentiation into dendrites [35,38]. Besides JNK, a member of another MAP kinase subfamily, p44/p42 extracellular signal-regulated kinase (ERK1/2), has also been implicated in neurite outgrowth and in the development of neuronal polarity [35,39].

During axonogenesis, JNK seems to be placed to regulate axonal growth directly, via its effects on the cytoskeleton, and by regulating transcription over the long term. Indeed, as reported above, JNK phosphorylates a variety of nuclear and cytoplasmic targets, including transcription factors, microtubule-associated proteins, and proteins directly interacting with the actin cytoskeleton [30]. Thus, the interaction between JNK and the cytoskeleton-related proteins plays a significant role in microtubule and actin dynamics for channelling the growth of the emerging axon. For instance, one substrate related to JNK pathway is the cyclic AMP-dependent transcription factor ATF-2, a member of the ATF/CREB (cAMP response element-binding protein) family of transcription factors that bind to CRE (cAMP-responsive element) consensus sites [40]. Once phosphorylated by JNK, ATF-2 is likely to lead to transcriptional changes that provide the materials necessary for axonal elongation and maintenance. In fact, the co-expression of c-Jun (JNK phosphorylation substrate) and active ATF-2 is increased in those retinal ganglion cells that survive after partial optic nerve injury and restore or partially maintain axonal connection after axotomy. Phospho-ATF-2 is enriched in the axon, similarly to phospho-JNK. Chronic or acute treatment with inhibitors decreases phospho-ATF-2, without affecting total ATF-2 levels. Thus, JNK is able to regulate axonal growth directly, via its effects on the cytoskeleton, and over the long term, by regulating transcription [41] (Figure 2B). In *Xenopus laevis* embryos, Hutchins and Szaro identified a post-transcriptional mechanism in which JNK regulates axonogenesis in spinal motor axons by phosphorylating a specific site on heterogeneous nuclear ribonucleoprotein K (hnRNP K). JNK phosphorylation of hnRNP K occurs within the cytoplasm and through phosphorylation of hnRNP K, JNKs posttranscriptionally regulate cytoskeletal genes that build the axon (e.g., type IV NFs, tau, ARP2, and GAP43). Indeed, JNK phosphorylation of hnRNP K affected neither nucleocytoplasmic localization nor RNA binding of hnRNP K but rather mediated the interaction between hnRNP K and the molecular machinery for translating its targeted RNAs, contributing to regulate translation of proteins crucial for axonogenesis. Both JNK inhibition and hnRNP K knockdown inhibited motor axon outgrowth and translation of hnRNP K-regulated cytoskeletal RNAs (tau and neurofilament medium) [38]. Finally, molecules able to phosphorylate JNK are involved in axonal growth. For instance, the peroxisome proliferator-activated receptor-γ (PPARγ) is able to stimulate axonal growth and accelerate neuronal polarity on different neuronal cell types via activation of JNK proteins. Indeed, PPARγ is selectively activated by thiazolidinediones (TZDs) drug, leading to the induced activation of the JNK pathway and promoting axonal elongation. The pharmacological blockage of this pathway prevents axon elongation induced by TZDs [42].

## 3. JNK in Developing Spinal Cord

After leaving the cell body, the axons manage to find their way following very precise—sometimes really long—paths in the nervous system, to reach their proper targets. Together with their role in regulating axonal outgrowth, JNK proteins are also required for axon guidance during CNS development. Indeed, it has been demonstrated that JNK molecules are activated by different guidance cues involved in axon pathfinding of different neuronal cell types [32,43], and required for maintaining the correct axonal trajectories, probably by both phosphorylating cytoskeletal targets and regulating gene expression [18]. Mice presenting genetic deletion of JNK-pathway components show axonal anomalies. For instance, the anterior commissure of Jnk1knockout mice degenerates starting from postnatal day (P)6 to P12, demonstrating that JNK1 is required for axon maintenance [10]. In the spinal cord JNK is also involved in the regulation of axonal pathfinding processes of different neuronal subtypes (Figure 2C).

### 3.1. JNK in Commissural Axons

In the developing spinal cord, the subpopulation of commissural axons projects firstly along the dorso-ventral axis towards the ventral midline and turns either anteriorly or posteriorly along the longitudinal axis after crossing the floor plate to the contralateral side. As early as E11, JNK regulates axonal pathfinding in spinal commissural neuron. The floor plate region is a source of both chemo-attractant and repellent cues, such as Netrin-1 and Slits [44], that are able to modulate the axon guidance process. According to the current models, the cell sensitivity to repellents released by the floor plate is silenced in those commissural growth cones that are pre-crossing the midline, while is switched-on in the post-crossing commissural growth cones, once expelled from the midline, in order to avoid them to re-crossing the floor plate. Post-crossing axons are then following the rostro-caudal gradients of guidance cues and turn caudally or rostrally in the lateral and ventral funiculi respectively [44,45].

JNK pathway is, indeed, required for netrin signaling in the developing nervous system. The guidance cue Netrin-1 increases JNK1—not JNK2 or JNK3—activity to enable midline crossing through a mechanism involving the cell adhesion receptors DCC (deleted in colorectal cancer) or DSCAM (Down’s syndrome cell adhesion molecule), the main mediators of netrin attraction in the nervous system [43,46]. In the developing spinal cord, phospho-JNK is highly expressed in commissural axons before and as they cross the floor plate, and Netrin-1 stimulation dramatically increases the level of endogenous phospho-JNK in commissural axon growth cones. In vivo, inhibition of JNK signaling by selective pharmacological inhibitors (e.g., SP600125) or JNK1 RNA interference (RNAi) is able to suppress Netrin-1-induced axon attraction. JNK1 knockdown during embryogenesis (by chicken in ovo electroporation) leads to defects in spinal cord commissural axon projection and pathfinding; while in vitro expression of the JNK1 construct rescues the defects of JNK1 knockdown on Netrin-1-induced axon outgrowth and attraction. The knockdown of JNK2 and JNK3 does not affect Netrin-1-induced neurite outgrowth and pathfinding, confirming that the mechanism is mediated only by JNK1 isoform [43]. Either Netrin-1 knockout or DCC-knockout mice show a defective projection of commissural axons [47,48]. In vitro, Netrin-1 increases endogenous JNK activity in primary neurons and its effect is inhibited either by JNK-, JNK signaling-, and DCC-function inhibitors (anti-DCC function-blocking antibody). The knockdown of DSCAM protein partially inhibits Netrin-1-induced JNK activation, while, if combined with the anti-DCC function blocking antibody, leads to a total abolishment of the Netrin-1 effect on JNK activity; thus, DCC cooperates with DSCAM through regulating JNK activity in Netrin signaling [43].

It has been demonstrated that the axon formation in neurons depends also on stage-specific regulation of microtubule stability by the dual leucine zipper kinase (DLK), a MAPKKK for JNK. Together with JNK, the DLK-JNK1 pathway facilitates axon formation, plausibly involving also Netrin-1 signaling [43,49].

Moreover, during floor plate crossing, commissural growth cones gain Wnt-Frizzled (glycoproteins secreted by the ventral midline cells in the floor plate and expressed in an antero-posterior gradient) signaling responsiveness, which is required for anterior turning of the most dorsal populations of post-crossing commissural axons [45]. Wnt-Frizzled signaling plays a fundamental role in spinal cord development and function by activating several pathways [50]. Among those, the highly conserved planar cell polarity (PCP) pathway has been demonstrated to be involved in Wnt-mediated axon guidance because of its ability to regulate the polarized cellular and tissue morphology in response to environmental cues [46,50]. PCP signaling is expressed in commissural axons and leads to the activation of JNK and c-Jun by phosphorylation [46,51]. Indeed, in commissural axons, Shaffer and colleagues demonstrated a co-immunoreactivity between phosphor-JNK and TAG-1, a component of the PCP pathway expressed in pre-crossing and crossing segments of the spinal cord. Phospho-JNK is especially enriched in the post-crossing segment in a spatio-temporal way (starting from E11). Wnt5a stimulation in vitro and in “open-book” preparation assays leads to PCP and consequent JNK activation, resulting in the regulation of spinal cord commissural axon guidance in the antero-posterior direction at the proper stage when commissural axons are making anterior turning decisions (E13 in rat). Inhibiting JNK activity leads to antero-posterior randomization of commissural axon trajectories of more than 60% of commissural axons, especially after midline crossing [46]. Therefore, JNK plays as a downstream effector of PCP signaling in regulating the antero-posterior guidance of commissural axons.

### 3.2. JNK in Motor Neuron Development

In the mouse spinal cord, MNs are generated around E9.5 from ventral domain progenitors of the ventricular zone [52]. MN cell bodies are prevented to leave the spinal cord by so-called boundary cap cells, which surround nerve roots and express repulsive factors [44], whereas MN axons extend from the floor plate to reach the proper exit point, where they go through the basal lamina and leave the spinal cord ventrally, to innervate the skeletal muscles [53]. The cellular and molecular mechanisms regulating the motor axon to breach the central-peripheral nervous system boundary, together with the capability of motor axons to reach the proper muscle targets, are still elusive. A spatial relationship exists between the position of MN cell body in the spinal cord and the trajectory of the axonal projection to peripheral muscle target [54]; moreover, newborn MNs express a variety of receptors and signaling molecules, which provide guidance cues for MN axons, including the Netrin-1 receptors DCC. Furthermore, there are some insight into the mechanisms of axonal rostral turning, which involves Wnt proteins and SHH [44]. Thus, giving that at least some molecular pattern expression in developing MN axons are shared with the commissural axons, we could speculate that also in this cell population, JNK phosphorylation is mediated by Netrin-1 and/or Wnt signaling to control MN axon pathfinding within and outside the spinal cord, acting at different multiple levels and by an intertwined manner. For instance, DCC and Netrin-1 knockout mice show defects in MN pathfinding: the exit point of the axons of ventral MNs is shifted dorsally, indicating that Netrin-1 normally guides axonal elongation ventrally [44,55].

### 3.3. JNK in Spinal Cord Patterning

Interestingly, the JNK pathway is involved in brain final patterning and neuronal cell number regulation. The double full knockout mice for Jnk1 and Jnk2 show a failure to complete neural tube closure, displaying an overt exencephaly phenotype, together with region-specific alterations in cell survival, with decreased apoptosis in the hindbrain neuroepithelium and increased apoptosis in the forebrain [18,56].

During brain development, programmed cell death (PCD) occurs as a controlled mechanism for the maintenance of the homeostasis and tissue integrity. The activation of JNK-c-Jun signaling is a crucial event for the PCD of neurons, including MNs; indeed, it has been shown that JNK-mediated c-Jun phosphorylation is a transient and reversible event that occurs when a MN is more dependent on survival signals (e.g., neurotrophic factors) than in the adult.

In vitro, the overexpression of c-Jun is able to induce neuronal cell death, that can be prevented by the dominant-negative Jun; moreover, the co-expression of the dominant negative-Jun with the JNK- blocking JBD domain of JIP1 is involved in the same pathway in JNK/c-Jun mediated cell death.

Rescue from PCD by deletion of pro-apoptotic genes, such as the *Bax* gene, leads to inhibition of c-Jun signaling and subsequent reduction in MN death. In vivo, data confirm the necessary, but not sufficient, involvement of JNK downstream protein c-Jun in the commitment to die: the phosphorylation of c-Jun was observed in MNs during embryonic PCD, and actions increasing cell survival (e.g., neuromuscular activity blockade) reduce the number of the phospho-c-Jun positive MNs [57,58]. Thus, in the embryo, JNK-c-Jun cascade seems to be important for the initiation of early and reversible mechanisms of PCD, if the survival-promoting signals are not sufficient or lacking [59]. Pharmacological inhibition of c-Jun upstream inducers- such as JNK and p38 kinases- also promotes the survival of embryonic motor, sensory, and sympathetic neurons [58,60,61]. Thus, JNKs are functional in various stages of spinal cord development and are required in different cell populations to provide the proper axonal position and PCD.

The dual role for JNK in CNS development, in promoting the outgrowth and differentiation of cellular components, together with regulating cell number via programmed cell death, has been observed also in the adult nervous system. Thus, the second part of the review will highlight the main functions of JNK pathway in both regeneration and degeneration conditions in adult spinal cord.

## 4. JNK in Adult CNS

### JNK in Axonal Degeneration and Regeneration

Among neuronal compartments, axons are highly specialized structures that require integrity and maintenance throughout their entire lifespan. After development, the normal functions of central and peripheral axons can be compromised by various insults such as trauma, demyelination, blockade of axonal transport, chemical toxicity as well as diseases, leading to progressive morphological and molecular changes in response to stress (e.g., swelling of the cell body and fragmentation of rough endoplasmic reticulum). As a consequence of axonal lesion (axotomy), the neuron triggers its axonal self-destruction program, probably due to the loss of maintenance factors, which result in cytoskeletal disassembly and granular degeneration of the axon distal to the injury site, toward the cell body [62].

Depending on the context, axonal injury can result in neuronal apoptosis, axonal degeneration and/or regeneration, and JNK is required for triggering each of these different responses (Figure 3).

In adulthood, neuronal death is an evident and rapid event after axotomy of central “intrinsic” neurons (neurons projecting within the CNS districts) and less prominent and delayed following axotomy of those central “extrinsic” cells whose axons project into the periphery (e.g., MN axons and cranial nerves) [19].

It has been demonstrated that JNK pathway triggers neuronal degeneration and death in different brain pathological conditions and diseases, such as ischemia [63], epilepsy [64], and Alzheimer’s disease [65], strongly suggesting the pro-apoptotic role of JNK pathway also after axon injury. For instance, alterations of JNK and p38 signaling pathways have been observed in vitro and in vivo as potentially associated with pathogenesis and neuronal apoptosis in Alzheimer’s disease [66].

Crushing or cutting the peripheral nerves represent the experimental model of choice to study the cellular and molecular mechanisms of axotomy-induced neuronal death and reinnervation. Indeed, it has been demonstrated that degeneration both after nerve injuries and axon pruning share several molecular pathways [67]. Among those molecules, JNK activation is associated with both neuronal death and regeneration, showing its dichotomy.

In adult CNS, indeed, the JNKs are involved in triggering cell death in different cell types and the expression of dominant negative inhibitors of nuclear JNK confers resistance to the apoptotic events following trophic support. JNK can modulate death signaling through the canonical phosphorylation of c-Jun pathway: it is of note that the non-phosphorylable cell mutants of c-Jun are resistant to apoptosis, and c-Jun deficiency results in resistance to axotomy-induced death [19]. Moreover, alternative pathways mediate JNKs contribution to cell death: some of them involve the phosphorylation of the p53 family proteins [14], others involve different cytoplasmic (e.g., Bcl-2, Bcl-X, and Bim) and nuclear substrates (e.g., ATF2; and Elk-1) that mediate JNK effects [68,69].

The extent of involvement of c-Jun or other nuclear/cytoplasmatic molecules in JNK-mediated regulation of axotomy-induced cell death still remains to be elucidated (Figure 3A). JNK signaling recruitment changes depending on cell types and conditions, as neuronal different populations can exhibit distinct temporal differences in the regulation of cell death, highlighting that the vulnerability of neuronal subtypes is specific. Moreover, the three isoforms and substrate-specific functions of JNK molecule add an additional level of complexity [69]. Indeed, JNK role in pro-apoptotic mechanisms is mediated by both cell type and time, as a lack of JNK2 and JNK3 isoforms in facial MNs significantly reduces their axotomy-induced death in neonatal mice, whereas death of adult facial MNs seems independent by JNKs [70,71], suggesting a developmental modulation of JNK actions in these neurons.

In other cell types—as nigrostriatal dopaminergic cells of the medial forebrain bundle—JNK promotes neuronal cell death mainly by phosphorylation of c-Jun downstream protein. Indeed, axotomy leads to a JNK3-dependent death of nigral neurons, and JNK3-deficient mice show enhanced and lasting survival of the axotomized dopaminergic neurons in the substantia nigra pars compacta [70]. In Purkinje cells, for example, c-Jun deficiency results in resistance to axotomy-induced death, while the overexpression of c-Jun enhances the death of injured cells [72], thus suggesting also a cell type-specific function of JNK-c-Jun mechanisms in both repair and death events.

The stress-activated MAPK pathways are also involved in the transmission of apoptotic signals following traumatic spinal cord injury (SCI): for instance, SCI leads to early ASK1 activation in both white and grey matters and subsequent phosphorylation of JNK and p38 MAPK observed in apoptotic cells [73]. Indeed, JNK contributes to trauma-induced DP5 expression and subsequent neuronal apoptosis mediated by c-Jun phosphorylation, and in vivo inhibition of JNK activity attenuates DP5 induction and caspase-3 activation, resulting in cell protection [74]. Moreover, JNK inhibition by peptide administration shows neuroprotective effects by preventing c-Jun phosphorylation and caspase-3 cleavage, resulting in an increased sparing of white matter at the spinal lesion site, pointing out that JNK inhibition could prevent the evolution of secondary damage after spinal cord injury [75]. Thus, JNK has been identified as a key element responsible for the regulation of apoptotic signals and therefore, it is critical for pathological cell death also following SCI.

Oxidative stress is also responsible of MN death by apoptosis in the spinal cord. Oxidative stress occurs when the production of the oxidizing agents overwhelms the normal antioxidant capacity system of the cells. This is detrimental to MNs and finally leads to neuronal cell death due to the reactive oxygen species (ROS) accumulation and mitochondrial dysfunctions (mitochondrial uncoupling and membrane depolarization). Indeed, after spinal cord injury, MN death could occur by oxidative stress through cytochrome C release [76]. It has been reported that oxidative stress in MNs in vitro induces the increase in expression levels of phosphorylated JNK proteins [76], and decreases the activation of other kinase molecules (such as ERK1/2) reported to promote cell survival [77].

Pharmacological intervention with molecules, which protect cells from oxidative stress damage—e.g., Midazolam—decreases the expression levels of phospho-JNK in MN cell lines while it leads to the increasing expression levels of anti-apoptotic kinases, promoting cell survival [78]. Other papers highlight the dual role of JNK pathway in promoting both MN degeneration and regeneration. Indeed it has been shown that in MN depleted of trophic factors, JNK inhibition attenuates caspase activation, nuclear condensation, and reduces cell death. Conversely, pharmacological JNK inhibition in healthy MNs supplied with trophic factors results in a decreased mitochondrial membrane potential, decreased phosphorylation of cytoskeletal MAP1 proteins, leading to cell degeneration [69].

JNK signaling is also involved in Wallerian degeneration, which occurs in the neuron at the site of the lesion without affecting the cell body viability, in absence of neuronal death [79,80] (Figure 3B). The axon degenerates in 3–4 days through a process of swelling and subsequent granulation, together with myelin. JNKs play a central role in Wallerian degeneration and subsequent regeneration. The JNK pathway is required for regulating axonal degeneration early after injury, before axon fragmentation. Indeed, the inhibition of JNK soon after injury is able to delay the ongoing degeneration, whereas inhibition starting during axonal fragmentation phase shows no effect [81]. The molecular mechanisms mediating JNK regulation of Wallerian degeneration remain partially unknown. JNK signaling after injury requires the orphan receptor death receptor 6 (DR6) to drive axon degeneration by further activation of calpains, as observed both in vitro and in vivo [82]. JNK phosphorylation after axonal damage dramatically decreases SCG10 affinity for tubulin, altering the balance between microtubule assembly and disassembly, and leading to SCG10 depletion and accelerated axonal degradation [83,84]. The pro-degenerative signaling promoted by JNK in axons can be blocked by the Wallerian degeneration slow (Wld^S^) protein activation, whose function is carried out through chaperone activity, at least in *Drosophila* models of axonal injury [85].

Axotomy leads to a fast and lasting JNK accumulation in the axotomized perikaryon (JNKs are retrogradely transported from the site of injury to the perikaryon), also characterized by the persistent phosphorylation of c-Jun in its N-terminal, which is a cellular response conserved in neurons. The accumulation of JNK might label those neurons prone to die. Inversely, the presence of activated JNK in axonal sprouting indicates that the pathway is involved in regenerative efforts to reconnect the neuron-target axis. Indeed, JNK activation is sustained during regeneration and return to basal levels once the process is completed [19] (Figure 3C). For instance, the Jnk1 knockout mouse shows a delay in axonal regeneration after facial nerve cut [86].

Interestingly, not all the three JNK isoforms play the same role in regeneration. Indeed, loss of JNK2 and JNK3 (and not JNK1) delays neuritogenesis in vitro and in vivo, suggesting their specific roles in either initiation or extension of regenerating neurites. JNK signaling is further required for promoting neurite elongation, as pharmacological JNK inhibition results in neurite retraction; this activity relies on JNK1 and JNK2 and not on JNK3. Moreover, in the cytoplasm, JNK1 and JNK2—but not JNK3—regulate the phosphorylation state of the microtubule-associated protein MAP1B, whose role in guiding the directionality of growth cone migration and axonal branching in regeneration has been proved. Indeed, a lack of MAP1B prevents neurite retraction induced by JNK inhibition [87,88]. Thus, different JNK isoforms show different implications in cytoskeletal reorganization leading to axon regeneration.

In the cell body, JNKs trigger both expression and activation of c-Jun downstream molecules; indeed, c-Jun is essential for triggering JNK effects on regeneration. In the CNS, the c-Jun depletion is correlated with the failure in axonal pruning, whereas its presence is related to the regrowth of axotomized neurons following spinal cord hemisection. Thus, the available data highlight that most of the neurons that survive axotomy and maintain the competence to regenerate the axons, also exhibit an early and lasting expression of c-Jun [19,89,90].

A further mechanism contributing to axonal regeneration is the stimulation of the receptors for cytokines or growth factors. Interestingly, at least in *Caenorhabditis elegans*, it has been demonstrated that, after axonal injury, the binding between the growth factor SVH-1 and its receptor tyrosine kinase, SVH-2, modulates the activation of the KGB-1 JNK-MAPK cascade to promote regeneration. Loss or deficiency in SVH-1-SVH-2 signaling leads to the impairment of neurons in regeneration capability. This event, moreover, seems to be peculiarly related to neuron regeneration processes, whereas it is not involved in neuronal development [91].

Apart from nuclear c-Jun, JNK proteins can phosphorylate also different proteins in the cytoplasm, like neurofilaments and STAT3, to regulate the axonal growth [19,92,93]. For instance, JNKs and STAT3 show a similar activity pattern, and STAT3 plays a similar role of JNK in regeneration after axotomy, as MNs of STAT3-knockout mice are more likely to undergo cell death than those of wild-type mice, implying a cooperation between JNK and STAT3 proteins to promote axonal outgrowth [94,95].

Determining the factors contributing to the dual role of JNK as a regulator of both regeneration and neuronal death remains an open question. Even though all JNK proteins are involved in neuritogenesis, whereas JNK2 and JNK3—and not JNK1 isoform—are more likely to mediate programmed cell death, it is difficult to address if a specific JNK isoform displays a pivotal role in degeneration and/or regeneration: indeed, the majority of experiments related to axotomy are performed in vitro, in order to easily gain insight into axon severing and into signal transduction mechanisms; however, this leads to a limited view of all the intrinsic cellular “states” of a neuron and the extracellular molecular environment that could be present in the brain parenchyma after axonal damage. Cytoskeletal alterations and local changes due to axonal cut could promote local JNK activation: JNK could be then involved in either the phosphorylation of cytoplasmic proteins for promoting axonal regrowth, or in the activation of genes in the nucleus for activating cell death signaling [96]. Thus, it is presumable that the Janus-faced characteristic of JNK in the brain depends on JNK upstream activators and/or downstream effectors that could steer JNK signaling in the regenerative/degenerative directions. For instance, p38 and JNK pathways are able to promote axon regeneration when they are coordinately activated, and axonal regeneration fails if the activity of either pathway is absent. The synchronized regulation of these two MAPK pathways is mediated by the E3 ubiquitin ligase Phr1 via selective and targeted degradation of the MAPKKKs DLK-1 and MLK-1 and by the MAPK phosphatase MKP7, which downregulates both p38 and JNK as observed in *C. elegans*, on JNK pathway molecule homologues [97]. The upstream cascades are also not so clearly understood. For instance, the dual leucine zipper kinase (DLK) seems to control the contradictory response of JNK signaling in mammalian neurons. Indeed, under certain circumstances, DLK interacts with JIP3 proteins and the mitogen-activated protein kinase kinase 7 (MKK7) to lead to JNK phosphorylation and trigger c-Jun and caspase signaling activation, to promote neurodegeneration and cell death. Conversely, DLK can activate MKK7 and MKK4 to promote JNK activation by phosphorylation in Wallerian degeneration processes. Indeed, after activation, JNK can bind SCG10 to increase axon degradation, and the involvement of JIP proteins in this mechanism remains to be elucidated. Finally, the DLK-dependent phosphorylation of JIP3 and MKK7 and MKK4 can also direct JNK activity towards specific neuronal responses (e.g., transcriptional activation of *ATF3, Sprr1a*, and *Klf6* genes), leading to axon regeneration [98,99]. To sum up, JNKs exert different functions in response to axonal injury, supporting both degeneration and regeneration. Understanding the upstream and downstream pathways involved in pro/anti-survival JNK recruitment would be of considerable importance in the perspective of acting on JNK pathway on one side to counteract degeneration and, on the other, to promote neuronal regeneration.

## 5. JNK in MN Degenerative Diseases

JNKs elicit neuronal death in a wide range of pathological contexts, characterized by axotomy or other type of cell excitotoxic insults, which lead to stress response activation in neurons [18]. Among these diseases, those affecting MNs are characterized by abnormalities of cytoskeletal regulation, especially at the axonal level, including neurofilament accumulation [100], oxidative stress sensitivity [101], and dysfunctions followed by a neurodegeneration induced by inflammatory agents and loss of trophic support [102]. All these insults can promote progressive cell death; moreover, evidence from animal models suggests that the neuronal dysfunctions precede the clinical phase of these diseases. Indeed, MN diseases are characterized by a partial degradation of nerve terminals and neuromuscular junctions at the early stages, while most cell bodies remain intact. MN pathology seems to initiate at the distal axon terminals—determining a loss of motor units and associated muscle functions—and to proceed in a “dying-back” pattern up to the cell body [103,104]. Therefore, one should focus on MN axons and terminals to study pathways involved in progressive MN degeneration, in order to delay or prevent neuronal degradation and death. Given the involvement of JNK molecules in degeneration and regeneration processes in neurons, this pathway is likely also involved in MN diseases. Thus, we will focus on JNK roles in three different pathologies affecting MNs: amyotrophic lateral sclerosis (ALS), commonly diagnosed in middle and late adulthood, spinal muscular atrophy (SMA), which occurs in early childhood (Figure 4), and the spinal and bulbar muscular atrophy, an adult-onset disorder characterized not only by the degeneration of spinal MNs, but also of those in the brainstem.

### 5.1. JNK Role in Amyotrophic Lateral Sclerosis

Amyotrophic lateral sclerosis (ALS) is a fatal neurodegenerative disease affecting both upper and lower MNs [105]. Approximately 90% of ALS cases are sporadic (sALS), whereas the remaining 10% are familiar (fALS) cases, with mutations identified in greater than 20 genes (including SOD1, TARDBP, FUS, and C9ORF72) across varies cellular functions; these mutations contribute for the two-third of fALS and for about the 10% of sALS [106]. However, sALS and fALS share a variety of complex and correlated pathological mechanisms that contribute to MN malfunction and death, including mitochondrial dysfunction, neuroinflammation, excitotoxicity, and protein aggregation [107]. This makes ALS a heterogeneous and multifactorial disease and explains the present lack of an effective treatment for all forms. Among the molecular mechanisms underlying ALS onset and progression, also JNK seems to play a role: given its function in cellular response to stress and activation of the canonical apoptotic pathway, the involvement of JNK signaling pathways in ALS disease seems to be ascertained (Figure 4A).

It has been shown that the mitogen-activated protein kinase MAP4K4 signaling results upregulated in stressed MNs derived from SOD1^G93A^ mice (the most common used ALS experimental model). MAP4K4 can be activated by TNFα and, in turn, it activates TAK1 and MKK4/MKK7 proteins, leading to the phosphorylation of JNK and, ultimately, c-Jun. Indeed, suppression of MAP4K4 by short interfering RNA (siRNA) silencing in both SOD1-mutant mice and human SOD1-mutant iPS-derived MNs improves cell survival and neurite outgrowth, indicating that the pro-MN survival by MAP4K4 inhibition is conserved across species. Additionally, these experiments highlighted that JNK3 (rather than JNK1 and 2) is the active isoform mediator of cell death via MAP4K4 activation. Indeed, using both MAP4K4 and JNK3 siRNA simultaneously do not lead to any increased survival compared with blocking either alone, suggesting that the two molecules are part of the same pathway in MN degeneration [108]. In stressed MNs, the upregulation of MAP4K4 leads to an increase in phospho-JNK and phospho-c-Jun levels, with the subsequent increase of cleaved caspase3 levels and activation of the canonical apoptotic pathway. Inhibition of MAP4K4 is able to reduce JNK and c-Jun phosphorylation levels and decrease cell degeneration. Knocking down c-Jun also leads to MN number rescue. Neuritic degeneration is an early event in neuronal disorders and often precedes MN death: SOD1-mutant MNs in vitro show reduced neuritic length and number of nodes per MN, manifesting signs of degeneration; inhibition of MAP4K4—and presumably of JNK—has a positive effect on neurite outgrowth, by maintaining its structure and preventing neurons from degeneration [108].

Interestingly, MAPK-JNK is also involved in the regulation of autophagy, as its phosphorylation causes a suppression of this critical intracellular pathway, which is essential for degrading aggregated and misfolded proteins in cells, to support survival [109]. Wu and colleagues indicated that autophagy is dysfunctional in ALS MNs, and MAP4K4 siRNA-mediated silencing induces a restoration of autophagic physiological rate, potentially contributing to the reduction observed in mutant SOD1. This mechanism seems not to be mediated by mTOR signaling, whereas it seems to be FoxO1 protein-dependent, as phosphorylated FoxO1 levels increase after MAP4K4 inhibition, consistent with findings that JNK deficiency increases FoxO1-mediated autophagy [108,110].

Together with MAP4K4, it has been proved that the dysregulation of transactive response DNA-binding protein-43 (TDP-43) is linked to ALS pathogenesis, as an overexpression level of the molecule has been found also in ALS patients. Increase in TDP-43 induces neuronal cell death through the upregulation of JNK levels. The expression of a dominant- negative JNK, the administration of JNK inhibitors, or dominant-negative c-Jun molecules reduces the TDP-43-mediated MN death in vitro [111].

Moreover, it has been proved that the activation of the dual leucine zipper kinase (DLK) signaling is detrimental in ALS, as well as in other neurodegenerative diseases like Alzheimer’s. Indeed, Le Pichons and colleagues (2017) confirmed an age-dependent increase in phosphorylation of JNK and c-Jun in neurons of the spinal cord (especially in lumbar portion) and in the cortex of SOD1^G93A^ and TDP-43^A315T^ mutated ALS models, as well as in lysates from sALS patients, confirming the activation of this pathway. By conditionally knocking-out the *DLK* gene, they revealed that DLK is required for the activation of JNK pathway and phosphorylation of c-Jun in neurons, as the removal of DLK is able to reduce the number of phospho-c-Jun positive cell in the spinal cord. DLK knock-out in the SOD1 model, interestingly, is able to significantly reduce MN cell death in vivo, from 40% neuronal loss in ALS mice of 14 weeks of age to a 13%. Moreover, by DLK inhibition a significant rescue in axon lumen area is observed in diseased mice, suggesting that the initial phase of axon degeneration is blocked or at least delayed. Thus, DLK is involved in JNK pathway activation and subsequent neuronal death. Acting on the DLK-JNK cascade could represent a promising neuroprotective approach: indeed, pharmacological inhibition of DLK (i.e., using GNE-8505 or GNA-3511 molecules) reduces the number of phospho-c-Jun positive cells in a dose dependent manner, delays neuromuscular junction denervation by almost 10% and increases MNs survival in the spinal cord (13% of neuronal lost instead of 40% [112]).

Other studies highlighted the involvement of JNK pathway in ALS disease. Deep RNA sequencing and gene expression profiling on iPS cell-based ALS models, followed by pharmacological screening, identified about 480 genes in the SOD1 mutation dataset that were differentially regulated, compared to control spinal cord. Among these molecules, ERK and JNK signaling have been highlighted as key drivers of neurodegeneration in SOD1 MN progenitors in vitro: indeed, the ERK/JNK downstream target AP1 complex member Jun (that has high-affinity for the AP-1 binding site in the Jun promoter region inducing c-Jun transcription) is highly expressed in MNs compared with non-MN cells, providing a mechanistic insight into the specific degeneration of MNs. Interestingly, this study revealed that the MAPK pathway is altered also in FUS mutated-MNs, suggesting that network perturbations induced by ALS converge partly on specific pathways [113].

Different strategies and pharmacological approaches designed to target JNKs and its upstream or downstream molecules have been tested on ALS models to reduce MN death. Together with pharmacological administration of molecular inhibitors, injections of viral genomes were used to easily pass the blood brain barrier (BBB) and efficiently deliver genes in ALS mutant mice: for instance, after delivering *IGF1* genes in SOD1^G93A^, the increase in IGF1-positive MNs is observed in the spinal cord, followed by an increase unbroken axon number, and reduction in neuroinflammation (astrogliosis) and in muscular atrophy. IGF1 acts directly on both the stress activated JNK and p38 kinase pathways, remarkably decreasing their phosphorylation [114]. Knowing the identity of those molecules upstream or downstream JNK that are dysregulated in ALS phenotypes may provide biochemical markers to enable earlier diagnosis of the pathology and molecular targets for developing new therapeutic compounds.

### 5.2. JNK Role in Spinal Muscular Atrophy

Spinal muscular atrophy (SMA) is an autosomal recessive MN disease, representing the most common genetic cause of infant mortality. SMA is due to mutation/deletion in the survival motor neuron 1 (*SMN1*) gene, and is characterized primarily by the degeneration of brainstem and spinal cord MNs, that leads to skeletal muscle progressive denervation and atrophy, followed by symmetric limb paralysis, respiratory failure, and finally death [115].

Indeed, the in vitro growth and differentiation of spinal cord MNs from SMA mice (SMNΔ7) results severely impaired compared with healthy MNs: neurons show growth defect and remarkable signs of neurodegeneration (axonal thickening, retraction, and swelling) [116,117]. SMN deficiency may result in intracellular stress that might activate intracellular signaling cascade and lead to neurodegeneration in SMA.

Genabai and colleagues reported that JNK is actually required for MN degeneration. They identified two MAPK signaling modules leading to full in vivo JNK activation in SMA: the cascades ASK1/MKK4/7/JNK and MEKK1/MKK4/7/JNK are both activated in the spinal cord of the SMA mouse model SMNΔ7 and patients—with increased phosphorylation of JNKs, glycogen synthase kinase 3 (GSK-3α and β), and ERK1—and mediate neuronal cell death caused by lower levels of SMN expression in MNs. Gemin5, a part of the SMN complex, interacts with ASK1, MKK4, and JNK, acting as a scaffold for the ASK1/MKK4/JNK signaling module to maintain the specificity of signaling. The difference in the levels of activation of MKK4 compared to MKK7 might suggest a tight regulation mechanism and specificity of the activation of the signaling modules [118,119].

The levels of phosphorylated JNK are 2.6 fold during postnatal development in SMA mouse spinal cord compared to wild-type condition, and in particular, among JNK isoforms, JNK3 is the one modulating neurodegeneration in SMA. Indeed, by genetically knocking-out JNK3 in SMA mice, the authors were able to perform a systemic rescue of the diseased phenotype, resulting in prevented neurodegeneration and death of spinal cord MNs, and in increased skeletal muscle innervation with subsequent reduction of atrophy and improvement of the overall growth Interestingly, the increase in JNK phosphorylation in SMA mice is accompanied by the increase in phospho-c-Jun levels, suggesting that the canonical apoptotic pathway (instead of other JNK-dependent pathways as ATF-2 recruitment) is activated in SMA condition [118]. Moreover, a similar activation of JNK-c-Jun-mediated cell death signaling cascade has been observed in a model of spinal muscular atrophy with respiratory distress type 1 (SMARD-1), under control of HG kinase [120].

Thus, these results suggest that JNK activation occurs early in SMA, and continues to increase with age, contributing to the increase in severity of the disease. JNK signaling pathway, by mediating neurodegeneration through c-Jun activation, might represent a potential SMN-independent target of intervention for the treatment of SMA (Figure 4B).

For this reason, recently, we pharmacologically inhibited the JNK cascade in SMA SMNΔ7 mice, by administering a cell-penetrating JNK inhibitor (named D-JNKI1) peptide of all the three isoforms [121]. This peptide has shown its efficacy in blocking JNK cascade and promoting neuroprotection in the nervous system in both in vivo and in vitro studies [64,65,122,123,124]. In contiguity with Genabai observations, in our work we confirmed that intracellular stress signaling pathway and apoptotic mechanisms are increased in the SMA spinal cord, as the phosphorylation of JNK pathway is higher in the later phase of SMA progression (postnatal age P12), compared with healthy mice of the same age. SMA mice show less MNs compared to wild-type; moreover, many phospho-c-Jun positive MNs are visible in SMA spinal ventral horns. The chronic pharmacological inhibition of JNK by D-JNKI1 is able to strongly reduce phospho-c-Jun levels in MNs, showing its protective role. Indeed, after DJNKI-1 treatment, a significant reduction of MN neurodegeneration, inflammation, and muscular atrophy is observed [125]. Thus, these data strongly support that, in SMN deficiency condition, JNK activation is implied in MN degeneration and subsequent muscular atrophy. The inhibition of JNK signaling pathway partially rescues SMA phenotype, by supporting the crosstalk between MNs and muscles, as already observed for instance after the inhibition of the ROCK pathway [126].

Recently, a novel mechanism underlying JNK-dependent SMA MN degeneration has been observed. Indeed, the zinc finger protein ZPR1 interacts with SMN protein and exerts a pro-survival effect on MNs: ZPR1 is important for the accumulation of SMN in the nucleus; SMA patients express lower levels of ZPR1 and the inhibition of the *Zpr1* gene in SMA mutant mice increases the severity of the pathology, with higher caspase-mediated neuron degeneration [127]. ZPR1-dependent neurodegeneration has been linked with JNK activity: it has been discovered that deficiency of ZPR1 in the SMA spinal cord leads to the activation of the MLK3/MKK7/JNK cascade, with the consequent phosphorylation of the c-Jun protein, followed by neuronal degeneration. This ZPR1-dependent neuron degeneration mechanism is specifically mediated by JNK3 activation, as in ZPR1-deficient degenerating neurons the MAPK array analysis shows only a small increase in JNK1 and JNK2 activation, in contrast with a marked increase (25-fold) in phospho-JNK3. Moreover, neurons derived from Jnk3-knockout mice show reduced degeneration (21%) compared with degeneration (75%) of wild-type neurons upon ZPR1 knockdown, confirming that loss of JNK3 markedly increases survival (almost 54%) of neurons. Moreover, genetic ablation of JNK3 and pharmacological JNK inhibition result in a reduction of phospho-c-Jun activity and reduced degeneration of ZPR1-deficient MNs. Therefore, prevention of ZPR1 loss may help in reducing JNK-mediated MN degeneration, and data confirm that JNK inhibition may be a viable therapeutic approach to delay or prevent MN degeneration in SMA [128].

Thus, our and other studies suggest that the pharmacological inhibition of JNK pathway could represent an effective potential adjuvant to other—already in use—SMN-dependent therapies for treating SMA disease (like the antisense oligonucleotides nusinersen, Biogen) [129] which constitute a very promising strategy for SMA, but still have some limitations [130].

However contrasting results come from other papers, which conversely report no differences in stress-activated JNK-c-Jun signaling in MNs of SMNΔ7 SMA mice and SMA human spinal cord compared to healthy tissue [131,132]. These inconsistencies can be ascribed to different time points considered for the analyses. JNK and/or other (pro-apoptotic) molecular pathways can be activated differently during the cell life: probably the analysis of their expression in MNs at later phases of the disease (e.g., P16 in delta7 mouse model, as in [131]) might not reveal any differences in SMA tissue, because at this stage the highest wave of MN degeneration has been already triggered and is concluding. However further studies are necessary to clarify these aspects.

### 5.3. JNK Role in Spinal and Bulbar Muscular Atrophy

Spinal bulbar muscular atrophy (SBMA) is a hereditary late-onset neurodegenerative disorder, which involves the expansion of the polyQ stretch in the androgen receptor (AR) [133]. SBMA is a lower MN disease, characterized by degeneration of cells in both spinal cord and brainstem, leading to proximal muscle weakness, muscle flaccidity, and atrophy [134]. This suggests that cellular processes particularly critical for MN survival and functions are selectively altered by polyQ-AR accumulation, even though the pathogenic mechanisms for SBMA remain unclear and no effective treatments are available.

Similarly to other MN diseases, JNK pathway is activated also in SBMA. Indeed, the accumulation of polyQ-AR neurotoxic fragments in neuronal cytoplasm results in caspase-dependent apoptotic cell death, which requires the activation of JNK pathway through phosphorylation of c-Jun [135,136].

The MNs affected in SBMA (located in the brainstem and spinal cord) have some of the longest axons in humans: such features render these cells particularly vulnerable to alteration in fast axonal transport (FAT). In vitro studies demonstrated that polyQ-AR expression induces FAT inhibition through a pathway involving JNK activation independently of hormone binding, the subsequent phosphorylation of kinesin-1 heavy chain (KHC) subunits by JNK, and the inhibition of kinesin-1 binding to microtubules [133], leading to progressive cell death.

Moreover, also JNK-dependent nuclear export of PolyQ-AR is impaired in SBMA neurons [137]. Although the molecular mechanism by which polyQ-AR increases JNK activity has not been well addressed yet, the increase in JNK activity in SBMA conditions, together with the inhibition of FAT by polyQ-AR expression, may represent a primary pathogenic event to produce this dying-back neuropathy. Thus, inhibiting JNK pathway to increase FAT in MNs could represent a favorable therapeutic target [133].

## 6. Conclusions and Perspectives

The concept to improve neuroprotection through inhibition of JNK isoforms is driving the search for specific drugs since many years. Despite that the efforts are in the directions to develop isoform-specific JNK inhibitors for CNS disorders, JNK inhibitors that do not discriminate well between JNK isoforms (e.g., DKJNI-1) demonstrate the most effective neuroprotection in vivo [138], consistent with the idea that all the JNK isoforms contribute to neuronal death. To selectively block the effect of JNK in the nucleus, the cytoplasm and the mitochondria, compartment-specific molecules against the pro-apoptotic JNK effects have been generated, by fusing a nuclear export sequence or a nuclear localization sequence upstream of the JNK inhibitor JBD domain. The nuclear localization sequence provides substantial protection from cell death, suggesting that the nucleus is the major cellular compartment in which JNK triggers death responses [27,139]. Thus, targeting JNK either in the nucleus or in mitochondria may confer neuroprotection and represent a promising therapeutic approach (reviewed in [18]).

JNK is an active key player in both the developing and adult brain, and exerts different roles depending on the cell compartment, age, and cellular condition. Alteration of JNK pathway during development leads to defects in axonal architecture and pathfinding; whereas in adulthood, the activation at several levels of JNK upstream and downstream pathways has been related to cell death, degeneration, and MN pathologies. However, a complete view of how JNKs coordinate all these events is lacking, and many new substrates of JNKs remain to be identified. Thus, understanding the mechanisms underlying all JNK functions would be extremely important, in order to act on these specific mechanisms to prevent cell death (as in case of neurodegenerative diseases as ALS, SMA, or SBMA), or, on the contrary, to promote axonal regeneration and cell integrity after injury.

## Figures and Tables

**Figure 1 cells-08-01576-f001:**
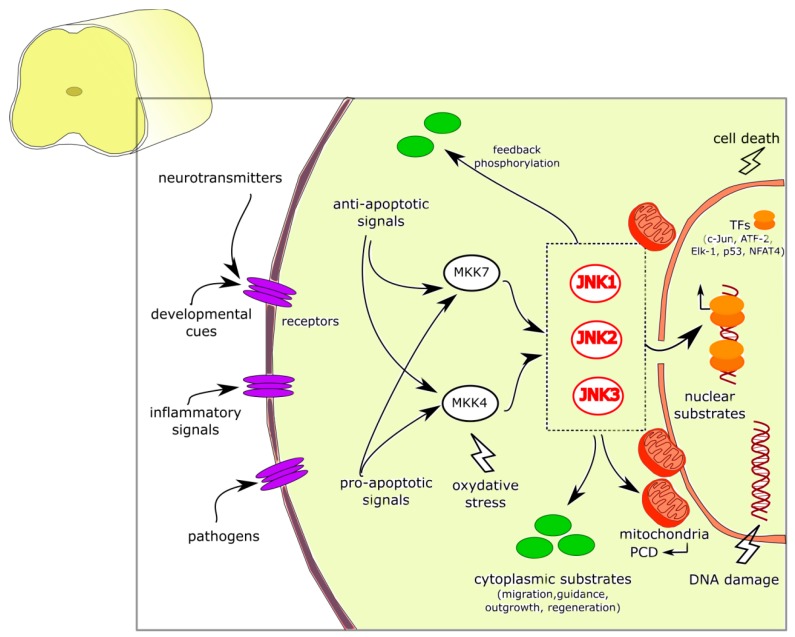
JNK signaling pathways and functions in the nucleus and cytoplasm of spinal cord neurons. JNK pathways are activated by different anti- or pro-apoptotic signals such as extracellular (e.g., inflammatory signals, pathogens, developmental factors, and neurotransmitters) as well as intracellular (e.g., oxidative stress, and DNA damage) stimuli that converge on the three JNK isoforms and promote JNK pleiotropic functions (physiological vs. stress-inducible): they can translocate in the nucleus and in mitochondria, or remain into the cytoplasm. JNKs phosphorylate a variety of cytoplasmic as well as nuclear substrates, can trigger the programmed cell death, promote a wide range of functional roles in brain development (e.g., cell migration, axonal guidance, neurite formation, and outgrowth) and in nerve regeneration. Abbreviations: JNK, c-Jun amino-terminal kinase; MKK4/7, mitogen-activated protein kinase kinase 4/7; PCD, programmed cell death, TFs, transcription factors.

**Figure 2 cells-08-01576-f002:**
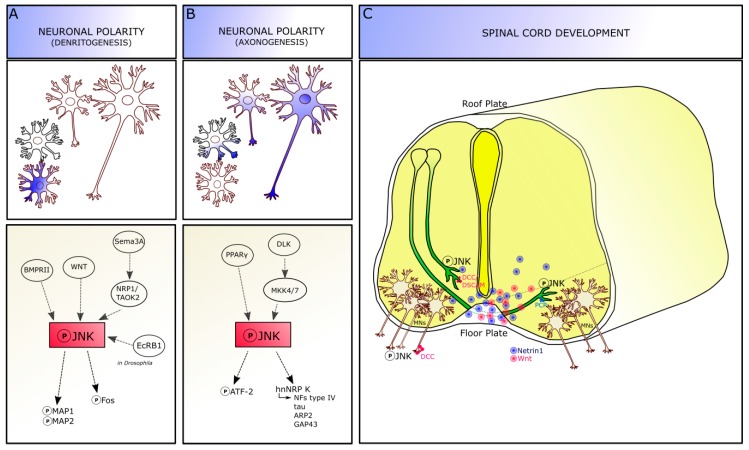
JNK roles in development. JNK signaling pathway is involved in axo-dendritic architecture formation and stabilization. Phosphorylation of JNK and its specific targets leads to dendrite (**A**) and axon (**B**) determination, thus contributing to neuronal polarity definition. (**C**) In developing spinal cord, JNK is required for axonal pathfinding of both commissural and motor axons. Abbreviations: JNK, c-Jun amino-terminal kinase; MAP, microtubule associated protein; Sema3a, semaphorin3a; BMPR, bone morphogenetic protein receptor; NRP, neuropilin; TAOK2, thousand-and one amino acid kinase 2; EcRB1, ecdysone receptor B1; DLK, dual leucine zipper kinase; MKK4/7, mitogen-activated protein kinase kinase 4/7; PPARγ, Peroxisome Proliferator-Activated Receptors-γ; ARP, actin-related protein; GAP43, growth associated protein 43; NF, neurofilament; DSCAM, down syndrome cell adhesion molecule; DCC, deleted in colorectal cancer; MNs, motor neurons.

**Figure 3 cells-08-01576-f003:**
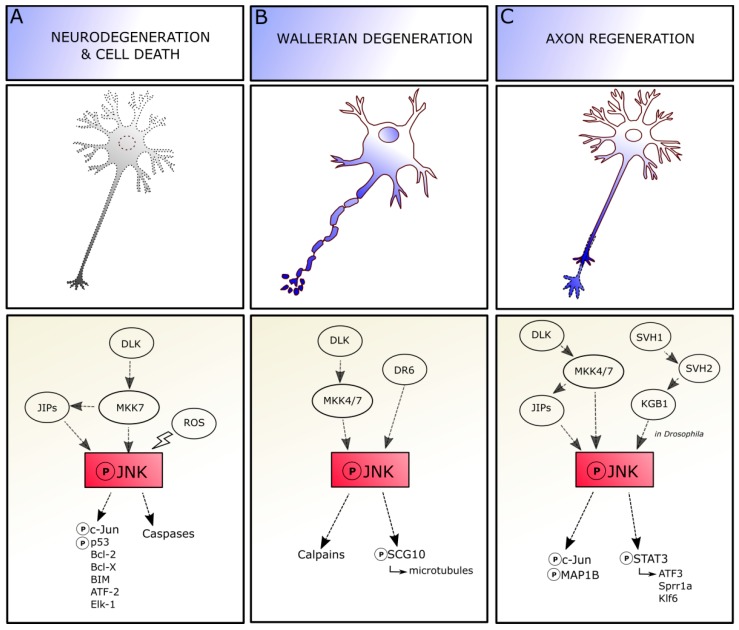
JNK pathways controlling contradictory responses in neurons. Depending on the context, axonal injury can result in neuronal cell death (**A**), axonal degeneration (**B**) and/or regeneration (**C**), and JNK is required for triggering each of these different responses. Abbreviations: JNK, c-Jun amino-terminal kinase; MKK4/7, mitogen-activated protein kinase kinase 4/7; DLK, dual leucine zipper kinase; JIP, JNK-interacting protein; ROS, reactive oxygen species; Bcl, B-cell lymphoma; ATF, activating transcription factor; MAP1B, microtubule associated protein 1B; DR6, death receptor 6; STAT, signal transducer and activator of transcription; Sprr, small proline rich proteins; Klf, Kruppel-like transcription factor.

**Figure 4 cells-08-01576-f004:**
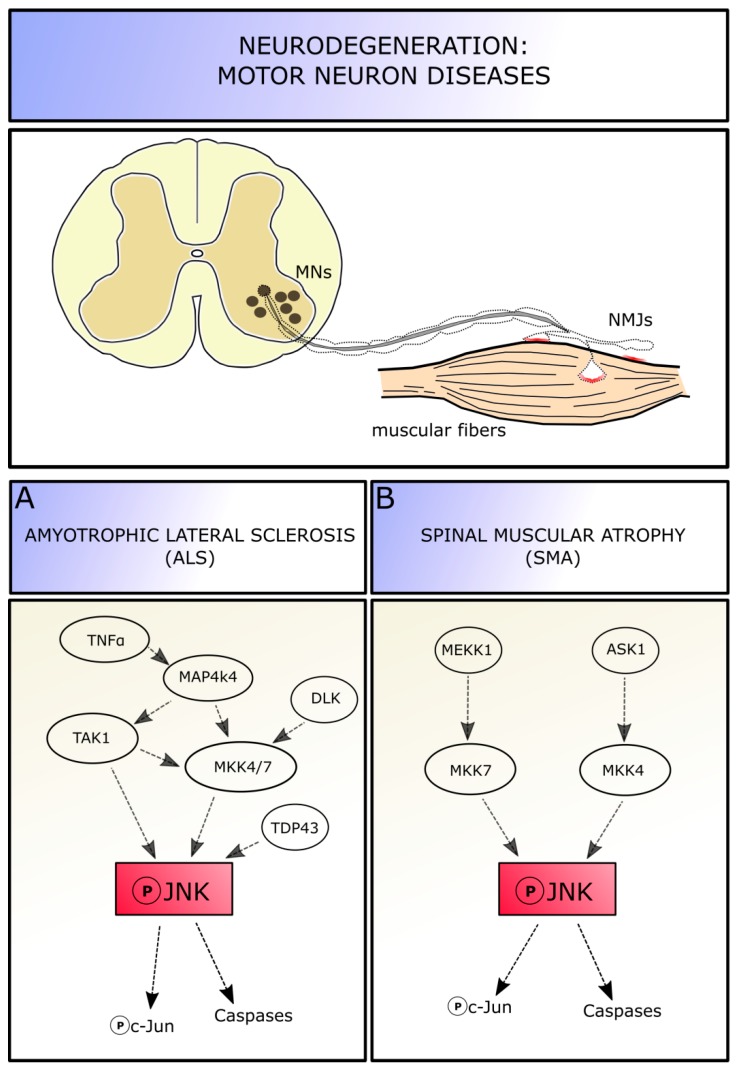
JNK involvement in motor neuron diseases. MN diseases are characterized by a partial degradation of nerve terminals and motor neuron junctions at the early stages. Within these MN pathologies, a role of JNK pathway has been demonstrated in amyotrophic lateral sclerosis (**A**) and spinal muscular atrophy (**B**). Abbreviations: MNs, motor neurons; NMJ, neuromuscular junctions; JNK, c-Jun amino-terminal kinase; MKK4/7, mitogen-activated protein kinase kinase 4/7; DLK, dual leucine zipper kinase; TNFα, tumor necrosis factor alpha; TAK1, transforming growth factor beta-activated kinase 1; MEKK-1, mitogen-activated protein kinase kinase kinase-1; ASK1, apoptosis signal-regulating kinase1.

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
