# Peer review of "JNK Signaling Pathway Involvement in Spinal Cord Neuron Development and Death"

_cells, 2019, doi:10.3390/cells8121576_

Round 1

Reviewer 1 Report

This review is interesting and well written. Figures are attractive and easily understandable as the key role of JNKs for the nervous system. The only thing that the appropriate figure, probably flow chart, for the component of Introduction, is hopefully required for the easier understanding of this study background.

Author Response

We thank the Rewiever for his/her constructive comments on the text. To satisfy the requests, we have added a new figure and its caption at the end of the Introduction section, to better explain JNK signaling pathway and protein localization in cells, and to make clearer the study background. We preferred to create a schematic illustration of JNK localization in spinal cord cells, instead of a flow chart, to standardize the style of this new image with that of the previous figures. Four figures are now present in the manuscript. The new figure is now named “Figure 1”, thus we have re-numbered the other following figures. The revisions we made on the text have been highlighted in red.

Reviewer 2 Report

This review (Schellino et al., JNK signaling pathway involved in spinal cord neuron development and death) aims to explain the rationale behind testing therapies based on inhibition of JNK signaling. It has been suggested that JNK3 expressed in the brain and activated by stress-stimuli, should be considered as a potential target for treating neurodegenerative mechanisms associated with AD, and other CNS diseases. This concept to improve neuroprotection via JNK inhibition is relatively new and will result in new drugs.

It is known that the c-Jun NH2-terminal protein kinase (JNK) is a Janus-faced kinase, which, in the nervous system, plays important roles in a broad range of physiological and pathological processes.

JNK family has been described as involved in injury responses and stress-59 induced apoptosis in neurodegenerative diseases and, more recently, in the pathophysiology of various neuropsychiatric disorders they are involved in the regulation of transcription of those genes 63 related to brain morphogenesis, together with those for axonal growth and pathfinding (Coffey Nat Rev Neurosci 2014 15: 285-99)

In this review, the authors summarize the involvement of JNK protein family in neuronal physiology and pathology, focusing on spinal cord.

The activation of JNK is required for axon formation. Indeed, JNK is expressed throughout the neurons, but its phosphorylated form is highly enriched in axons.

The main point is that treatment with specific JNK inhibitors are able to reversibly but selectively inhibit axonogenesis. During axonogenesis, JNK seems to be placed to regulate axonal growth directly, via its effects on the cytoskeleton, and by regulating transcription over the long term.

The following papers have to be involved in the Discussion in order to make this review more comprehensive.

Yin et al. (2005, Neurobiology of Disease 20: 881-889) ) observed that JNK activation contributes to trauma-induced DP% expression and subsequent apoptosis in spinal cord injury (SCI). Nakahara et al. (1999, Journal of Neuropathology and Experimental Neurology 58: 442-450) Numerous experimental studies have been performed to clarify the complex pathophysiology of spinal cord injury.
It has recently been reported the JNK and p38 signaling cascades may participate in signaling pathways leading to neuronal apoptosis. Yarza et al. (2016, Front Pharmacol. 6:321) JNK has been identified as a key element responsible for the regulation of apoptosis signals and therefore, it is critical for pathological cell death associated with neurodegenerative diseases. Repici et al. (Neurobiology of Disease 46: 710-721, 2012) have shown that specific JNK inhibition by a single injection of D-JNK11 prevents c-jun phosphorylation and caspase-3 cleavage, and has neuroprotective effects. The treatment results in an increased sparing of white matter at the lesion site in mouse spinal cord injured.

Otherwise, it is an important and well-written review.

Author Response

We thank the Reviewer fot his/her constructive comments on the text.

To satisfy the requests raised on the discussion part, we have introduced the papers suggested by the Reviewer in Chapter 4 “JNK in adult CNS - JNK in axonal degeneration and regeneration”(Lines 319-320 and lines 352-363). All the revisions we made on the text have been highlighted in red.

The suggested papers gave us the possibility to briefly introduce the involvement of JNK pathway also in spinal cord injury. Thus, we have now re-numbered the cited papers in the “References “ chapter, according to their new position in the text.